# Sutureless Purely Off-Clamp Robot-Assisted Partial Nephrectomy: Avoiding Renorrhaphy Does Not Jeopardize Surgical and Functional Outcomes

**DOI:** 10.3390/cancers15030698

**Published:** 2023-01-23

**Authors:** Aldo Brassetti, Leonardo Misuraca, Umberto Anceschi, Alfredo Maria Bove, Manuela Costantini, Maria Consiglia Ferriero, Salvatore Guaglianone, Riccardo Mastroianni, Giulia Torregiani, Marco Covotta, Gabriele Tuderti, Giuseppe Simone

**Affiliations:** 1Department of Urology, IRCCS “Regina Elena” National Cancer Institute, 00144 Rome, Italy; 2Department of Anesthesiology, IRCCS “Regina Elena” National Cancer Institute, 00144 Rome, Italy

**Keywords:** partial nephrectomy, renorrhaphy, sutureless, renal function, renal cancer

## Abstract

**Simple Summary:**

Suturing the kidney after tumor excision can be omitted most of the time, without increasing the risks of complications or jeopardizing renal function.

**Abstract:**

To compare outcomes of sutureless (SL) vs. renorrhaphy (RR) off-clamp robotic partial nephrectomy (ocRPN), we retrospectively analyzed procedures performed at our center, from January 2017 to April 2021, for cT1-2N0M0 renal masses. All the patients with a minimum follow-up < 1 month were excluded from the analysis. The trifecta rate defined surgical quality. Any worsening from chronic kidney disease (CKD) I-II to ≧ IIIa (from IIIa to ≧ IIIb, and from IIIb to ≧ IV) was considered as significant stage migration (sCKDsm). A 1:1 propensity score-matched (PSM) analysis minimized baseline imbalances between SL and RR cohorts in terms of age, gender, ASA score, baseline estimated glomerular filtration rate (eGFR), tumor size, and RENAL score. Logistic regression analyses identified predictors of trifecta achievement. Kaplan–Meier (KM) analysis assessed the impact of RR on significant chronic kidney disease sCKDsm-free survival (SMFS), while Cox regression analyses identified its predictors. Overall, 531 patients were included, with a median tumor size of 3.5 cm (IQR: 2.7–5); 70 (13%) presented with a cT2 mass. An SL approach was pursued in 180 cases, but 10 needed conversion to RR. After PSM analysis, patients receiving SL showed a higher trifecta rate (94% vs. 84%; *p* = 0.007). SMFS probabilities were comparable at KM analysis (log-rank = 0.69). Age (OR: 0.97; 95%CI: 0.95–0.99; *p* = 0.01), a RENAL score ≧ 10 (OR: 0.29; 95%CI: 0.15–0.57; *p* < 0.001), and RR (OR: 0.34; 95%CI: 0.17–0.67; *p* = 0.002) were independent predictors of trifecta achievement. Age (OR: 1.04; 95%CI: 1.003–1.07; *p* = 0.03) and baseline eGFR (OR: 0.99; 95%CI: 0.97–0.99; *p* = 0.05) independently predicted sCKDsm. Compared to RR, our experience seems to show that the SL approach significantly increased the probabilities of achieving the trifecta in the observed group of cases.

## 1. Introduction

Over the last decades, the improvement and spread of nephron-sparing surgery (NSS) techniques, together with the increasing trend in the use of robot-assisted partial nephrectomy (RAPN) for the treatment of renal masses, led to the adoption of radical nephrectomy for cT1 renal masses to less than 10%, especially in high volume centers [1].

However, despite the efforts to preserve long-term renal function and minimize the risk of acute kidney injury, up to 30% of patients with normal preoperative kidney function may experience a decline in estimated glomerular filtration rate (eGFR) to <60 mL/min/1.73 m^2^, and up to 10% may have a 50% reduction in renal function after surgery [2]. Thus, the search for innovative techniques to improve long-term functional outcomes after nephron-sparing surgery (NSS) is ongoing.

There is grounded evidence that every minute of warm ischemia significantly affects short- and long-term renal function [3]. Selective renal artery clamping [4], early unclamping [5], and off-clamp surgery [6] may help reduce the risk of postoperative renal function impairment. According to limited data, renorrhaphy (RR) may adversely affect functional outcomes, leading to ischemic necrosis of the stitched parenchyma [7,8] and potentially causing pseudoaneurysms and arteriovenous fistulas [9].

Since the pioneering era of minimally invasive PN, we aimed at minimizing ischemia “whenever feasible”; in 2012, we reported the feasibility and safety of *sutureless* [SL] laparoscopic PN for small exophytic renal tumors [10]. This experience shed light on the possibility of simplifying surgical procedures (in selected cases) and optimizing functional outcomes without jeopardizing oncologic results. However, the selection bias of that study limited the employment of the SL approach in intermediate/high nephrometry score renal tumors. In the last decade, we have witnessed the wide spreading of robotic platforms, and RAPN has overcome the laparoscopic approach for the treatment of all renal tumors. In this scenario, we developed a purely off-clamp SL robotic technique that completely avoids both hilar clamping and RR, based on the selective control of feeding arteries during tumor enucleation and the monopolar cauterization of the parenchymal breach to achieve hemostasis. In the present paper, we compare surgical and functional outcomes of patients who underwent purely off-clamp RAPN (ocRAPN), with or without RR, at our institution.

## 2. Materials and Methods

### 2.1. Patients and Dataset

After review board approval, our prospectively maintained database was queried for patients who underwent ocRAPN for organ-confined (cT1-2N0M0) renal tumors, from January 2017 onwards. Starting from January 2020, all the NSSs were performed with an SL approach; intraoperative conversion to RR was only deemed necessary in case of renal calyx violation with an obvious urinary leak and/or insufficient hemostasis at the first attempt of coagulation of the parenchymal breach.

The following data were extracted:Age, gender, and race;Baseline American Society of Anesthesiologists (ASA) score;Tumor side, clinical size, and surgical complexity (defined according to the R.E.N.A.L. score) [11];Hemostatic technique and eventual conversions from SL to RR;Serum creatinine levels assessed at baseline, at discharge, and 3, 6, and 12 months after surgery. For each timepoint, eGFR was calculated by means of the Chronic Kidney Disease Epidemiology Collaboration formula [12] and the National Kidney Foundation (NKF); chronic kidney disease (CKD) stages were defined accordingly [13]. Based on the NKF recommendations, a >30% reduction in the postoperative eGFR was considered as a “significant renal function deterioration” (sRFD), while any worsening from stages I-II to ≧ IIIa (from IIIa to ≧ IIIb, and from IIIb to ≧ IV) was defined as “significant CKD stage migration” (sCKDsm) [14,15];Postoperative complications (stratified according to the Clavien–Dindo (CD) classification system [16]) and length of hospital stay (LOS);Perioperative outcomes combined into our previously published trifecta (negative surgical margins, no CD≧3 complications, and no sRFD) to assess surgical quality [14].Final histology;Functional outcomes at last available follow-up.

### 2.2. Perioperative Care and Surgical Technique

All the procedures were performed by one single experienced robotic surgeon (G.S.). The surgical technique for conventional ocRPN with RR was previously described [17], and its principles remained unchanged, except for the postenucleation hemostasis. A single dose of cefazolin was given intravenously, 20 min before the skin incision. A transurethral urinary catheter was inserted, and the patient was placed in an extended flank position. The camera port was placed on the pararectal line, at the level of the umbilicus, and two other robotic cannulas were placed along the midclavicular and anterior axillary line, respectively. Two laparoscopic 12 mm trocars for the table-side assistant were positioned midway between the camera and each robotic port, thus creating a U-shape disposition focused on the tumor. The *daVinci Xi™* surgical platform (Intuitive Surgical, Sunnyvale, CA, USA), equipped with the *ERBE VIO dV 2.0™* energy generator, was docked from the patient’s back. Three robotic instruments were used: a 30° 3DHD endoscope, monopolar curved scissors, and *ProGrasp™* forceps (Intuitive Surgical). The two 12 mm assistant ports allowed the introduction of two suction-irrigation devices. The surgery was always carried out through a three-arm transperitoneal approach. The colon was medialized, and the Gerota’s fascia was opened, directly chasing the tumor. Dissection of the renal hilum was not attempted, as in no case were the vessels clamped. In fact, an hilum-sparing approach helps reduce surgical times while limiting accidental vascular injuries and, ultimately, intraoperative blood loss [18]. Straight access to the neoplasm was always pursued; thus, an extensive mobilization of the kidney was only required in the case of posterior tumors. Once the renal mass was identified, its margins were marked with cautery, intra-abdominal pressure was temporarily risen from 12 to 20 mmHg, and enucleation was begun. This step was carried out with blunt dissection, following an almost avascular plane, while the two suction-irrigation devices helped maintain a clear surgical field, combining irrigation and suction. The feeding arteries identified during the enucleation were progressively controlled with monopolar coagulation, but complete hemostasis was only pursued once the tumor was excised; then, monopolar cautery (*SwiftCOAG™* waveform, effect 8, with an 80 W limit) was extensively applied to the resection bed, under an appropriate dripping of saline solution, while blood was aspirated. Once a firm uniform eschar covered the entire resection bed and no bleeders were observed, the pneumoperitoneum was lowered down to 5 mmHg, and hemostasis was checked for 5 min. Topical hemostatic agents were never used to improve bleeding control. An EndoCatch™ device was used to retrieve the specimen.

Postoperative pain control was achieved using intravenous nonopioid analgesics, with a gradual transition to oral painkillers. On postoperative day (POD) 1, both the abdominal drain and urethral catheter were usually removed, oral intake was initiated, and patients were encouraged to ambulate. The drain was left in place when the 24 h output was >100 mL.

### 2.3. Study Objective

The aim of the present study was to present our novel SL surgical technique and report our single-center experience of 175 cases, retrospectively comparing outcomes with those of the conventional ocRAPN with RR.

### 2.4. Statistical Analysis

The study population was split into 2 groups according to hemostasis technique (RR vs. SL). After excluding 37 patients (7%) who underwent further renal surgeries because of bilateral tumors or renal stones, a 1:1 propensity score-matched (PSM) analysis was used to minimize imbalances between the two cohorts in variables potentially affecting outcomes (age, gender, ASA score, baseline eGFR, and RENAL score); the model was set to provide a standardized mean difference ≦10% between covariates. Frequencies and proportions were used to report categorical variables, which were compared by means of the χ^2^ test. Continuous variables were presented as median and interquartile ranges (IQRs) and were compared using either the Mann–Whitney U test or Kruskal–Wallis one-way analysis based on their normal or not-normal distribution, respectively (the normality of the distribution of variables was tested by the Kolmogorov–Smirnov test). Kaplan–Meier (KM) analysis and a log-rank test assessed the impact of surgical technique on sCKDsm-free survival (SMFS), while Cox regression analyses identified its predictors. Univariable and multivariable logistic regression analyses were performed to identify predictors of trifecta nonachievement and sRFI. The significance threshold was set at <0.05. Statistical analysis was performed using the Statistical Package for Social Science v. 25.0 (IBM, Somers, NY, USA).

## 3. Results

Out of 548 cases of ocRAPN, 15 were lost to follow-up and excluded from the analysis. Among the remaining 533 patients, 185 were offered an SL approach, but 10 (5%) required conversion to RR due to intraoperative suspicion of urinary calyx injury (*n* = 8) or uncontrollable bleeding (*n* = 2); therefore, they were ultimately allocated to the RR group (Appendix A).

Overall, the median age of our study population was 62 years (IQR: 54–71), and the average tumor size was 3.5 cm (IQR: 2.7–5); the observed trifecta rate was 86% (Table 1). Compared to those in the RR group (*n* = 358), patients from the SL cohort were significantly older (64 years vs. 61 years; *p* = 0.04) and less frequently harbored highly complex neoplasms (RENAL score ≥ 10) (17% vs. 23%; *p* = 0.005) (Table 1). Major complications (2% vs. 5%; *p* = 0.06) and transfusion (3% vs. 5%; *p* = 0.40) rates were not statistically different in the two groups. Overall, five (1%) urinary fistulas at the level of the parenchymal breach were observed after surgery (three and two in the SL and RR groups, respectively; *p* = 0.18), and they all healed spontaneously after ureteral stent placement. Selective embolization under general anesthesia was required 16 times overall because of post-RAPN bleeding.

Median follow-up time was 17 months (IQR: 8–30): 27 months (IQR: 14–36) in the RR cohort and 9 months (IQR: 6–14) in the SL group.

The PSM analysis generated two cohorts of 80 patients each, homogeneous for age, gender, ASA score, baseline eGFR, tumor size, and RENAL score (all *p* > 0.12). Patients receiving SL had shorter hospital stays (2 days vs. 3 days; *p* < 0.001) and increased likelihoods of achieving trifecta (96% vs. 84%; *p* = 0.008) (Table 1). SMFS probabilities were comparable at KM analysis (log-rank = 0.183) (Figure 1). Multivariable logistic regression analysis identified age (OR: 0.97; 95%CI: 0.95–0.99; *p* = 0.01), a RENAL score ≧ 10 (OR: 0.29; 95%CI: 0.15–0.57; *p* < 0.001), and renorrhaphy (OR: 0.34; 95%CI: 0.17–0.67; *p* = 0.002) as independent predictors of trifecta nonachievement (Table 2). Age at surgery (OR: 1.07; 95%CI: 1.04–1.11; *p* < 0.001), a RENAL score ≧ 10 (OR: 4.96; 95%CI: 2.23–11.06; *p* < 0.001), and RR (OR: 2.35; 95%CI: 1.12–4.94; *p* = 0.02) independently predicted sRFD (Appendix A). At Cox analyses, age (HR: 1.03; 95%CI: 1.003–1.07; *p* = 0.03) and baseline eGFR (HR: 0.99; 95%CI: 0.97–0.99; *p* = 0.05) independently predicted sCKDsm (Table 2).

## 4. Discussion

An average 20% decline in function after NSS is observed in the treated kidney right after partial nephrectomy. Such deterioration is mainly due to a resected healthy parenchymal margin, ischemia/reperfusion damage at the time of hilar clamping, and “reconstructive injury” caused by RR [19]. The first two variables have been extensively investigated, as these were considered the only modifiable factors. A recent multicenter prospective study on 507 NSSs provided evidence that both the resection technique (pure enucleation vs. enucleoresection) and warm ischemia time are independent predictors of postoperative acute kidney injury [20]. As “every minute counts when the renal hilum is clamped during partial nephrectomy” [3], different approaches to minimize hypoperfusion were proposed, such as preoperative superselective transarterial embolization [21], parenchymal [22] or superselective [23] clamping, early unclamping [24], zero ischemia [25], and a purely off-clamp approach [26]. The benefits of these techniques are particularly evident in selected clinical settings, such as patients who are candidates for imperative NSS [27], while for those with a normal eGFR at baseline, their advantage remains controversial.

As said above, the reconstructive injury caused by RR has been considered unavoidable for decades. This, however, is hypothesized to be responsible for 2/3 of the postoperative functional loss [7]. An interesting in vivo study showed that the depth of the necrosis of the renal parenchyma caused by RR ranges between 7 and 15 mm in pigs [8]. Hidas et al. provided evidence that omitting cortical stitching during open partial nephrectomy with cold ischemia provides an advantage in terms of function loss according to postoperative nuclear scans (−11.5% vs. −20.4%; *p* = 0.02) [28]. Similarly, Bahler and colleagues observed a significant volume preservation benefit (−3.8% vs. −15.6%; *p* < 0.001) in patients receiving SL on-clamp RAPN (*n* = 38) compared to those who received RR (*n* = 118); parenchymal suture continued to be an independent predictor of kidney damage according to multivariate analysis (*p* < 0.001) and after PSM analysis (*p* = 0.004) [29].

Since renorrhaphy affects the postoperative glomerular filtration rate and may even cause severe complications, such as pseudoaneurysms and arteriovenous fistulas [9,30], we hypothesized that achieving hemostasis through an SL approach could result in improved functional and surgical outcomes. Actually, electrocauterization has been already successfully employed for the same purposes during liver [31], lung [32], and pancreatic [33] surgeries. In fact, modern electrical scalpels are capable of delivering the proper amount of energy to the tissues so that Joule heat is produced locally, proteins coagulate at a temperature of 70–80 °C, and bleeding stops, while collateral carbonization is limited. A recent Japanese in vivo study tested the effects of electrocautery on a porcine renal model; delivering monopolar energy until hemostasis was achieved, and a 4.6 mm denaturation depth was observed, regardless of cauterization time. Interestingly, a lack or excess of saline solution dripping on the surgical field resulted in an average 2 mm penetration depth and insufficient bleeding control [34]. In 2014, Ota et al. published the first report on the use of coagulation during 39 laparoscopic partial nephrectomies (LPNs) (two-thirds were on-clamp NSSs); all operations were uneventful, and none required blood transfusions or conversion to nephrectomy. Eleven cases (28%), though, necessitated intraoperative repair of the collecting system. No significant decrease in renal function was observed within the study period [35]. Similar results were reported by Hongo et al. in a series of 32 off-clamp LPNs [36]. In 2019, Tohi and colleagues compared outcomes of SL on-clamp RAPN for cT1a (*n* = 66) and cT1b renal masses (*n* = 34); as expected, operation (154 vs. 184 min; *p* < 0.001) and ischemia times (14 vs. 21 min; *p* < 0.001) appeared significantly longer in the latter cohort, while the rate of positive surgical margin was comparable in the two groups (4.5% vs. 11.7%; *p* = 0.22). Of note, three major complications were observed, and all occurred in the cohort with larger tumors [37]. Two other small retrospective controlled studies showed a benefit in renal function and volume preservation when omitting cortical renorrhaphy during minimally invasive NSS [30,38].

In the present paper, we report outcomes of a continuous series of patients who underwent ocRAPN at our center. From January 2020, all those presenting with a cT1-2N0M0 renal mass that was suitable for ocRAPN were treated with an SL intent; 10/180 cases (5%) required RR because of an intraoperative suspicion of iatrogenic renal calyx injury (*n* = 8) or uncontrollable bleeding (*n* = 2). The latter presented with large (median tumor size: 4.7 cm; IQR: 3–6) and complex (60% had a RENAL score of 7–9; 40% had a RENAL score ≥ 10) neoplasms (Appendix A). Most of the patients in the SL group harbored small (3 cm; IQR: 2.5–5), uncomplex renal tumors (83% had a RENAL score <10), and less than one out of five presented with an ASA score ≧ 3. When compared to patients from the RR cohort (whose baseline characteristics were slightly, though not significantly, worse), the former reported a higher trifecta rate (93% vs. 83%; *p* < 0.001), and this difference was confirmed after PSM analysis (96% vs. 84%; *p* = 0.008) (Table 1). Interestingly, the multivariable logistic regression analysis confirmed that RR significantly reduces the probabilities of achieving trifecta (OR: 0.34; 95%CI: 0.17–0.67; *p* = 0.002), together with age (OR: 0.97; 95%CI: 0.95–0.99; *p* = 0.01) and a RENAL score ≧ 10 (OR: 0.29; 95%CI: 0.15–0.57; *p* < 0.001) (Table 2). Of note, the rate of sRFD was significantly reduced in the SL cohort (6% vs. 12%; *p* = 0.02) (Table 1), but this advantage, in terms of kidney function preservation, was only observed at hospital discharge and did not translate to a reduced risk of sCKD stage migration at a 12-month follow-up (Table 2, Figure 1). In fact, only age (HR: 1.03; 95%CI: 1.003–1.07; *p* = 0.03) and preoperative eGFR (HR: 0.99; 95%CI: 0.97–0.99; *p* = 0.05) independently predicted a significant worsening of renal function at Cox analysis.

Additionally, the rates of CD grade ≧3 complications (2% vs. 5%; *p* = 0.12), transfusions (3% vs. 4%; *p* = 0.62), and urinary fistulas (2% vs. 1%; *p* = 0.18) were comparable in the two groups, and were in line with other published series concerning ocRAPN [17]. These findings are of particular interest, considering that there are reports of urinomas due to coagulation in LPN [36,38], and it was consequently suggested to exclude from an SL approach all the patients presenting with a renal mass whose distance to the collecting system was < 5 mm; cautery, in fact, could slow the healing of heat-damaged tissues and make it harder to suture iatrogenic lesions of renal calyces [39].

It is worth highlighting that RR was an independent predictor of the risk of sRFD (OR: 2.35; 95%CI: 1.12–4.94; *p* = 0.02), together with age at surgery (OR: 1.07; 95%CI: 1.04–1.11; *p* < 0.001) and a RENAL score ≧ 10 (OR: 4.96; 95%CI: 2.23–11.06; *p* < 0.001). These results are in line with the available literature; in fact, it was hypothesized that the reconstructive injury caused by stitching the parenchymal breach may contribute to the postoperative functional loss [7]. Moreover, it is well-known that GFR decreases with age and that older patients have a reduced capability to cope with ischemic kidney damage [13]. Furthermore, there is strong evidence that highly complex tumors are associated with an increased contact surface area, which in turn requires more energy (in the case of an SL approach) or stitches (when RR is performed) to achieve hemostasis [11], thus increasing the risk of damaging the remaining healthy parenchyma. Surprisingly, our logistic regression analyses failed to prove any association between gender and the response to renal injury; in fact, there is evidence that women may be more protected against temporarily impaired renal circulation [40,41].

To the best of our knowledge, the series of RAPNs herein reported is the first ever published with an SL purely off-clamp approach. This study, however, suffers from the limitations inherent to its retrospective design; because of this, it should be considered as a hypothesis-generating study, and results from a randomized controlled trial are awaited in order to assess the possible advantages of any of the two techniques. Moreover, most of the patients in the SL group harbored small (3 cm; IQR: 2.5–5), uncomplex renal tumors (83% had a RENAL score <10), and less than one out of five presented with an ASA score ≧ 3; this significantly limits the generalizability of our findings. Furthermore, to evaluate the potential benefits of this procedure on long-term outcomes, a longer follow-up is required. Another limitation of the present study is that the two groups were not homogeneous in terms of the surgeon’s experience with the two techniques. In fact, in 2017, G.S. had already completed his learning curve with the conventional RR-ocRAPN; on the contrary, the SL approach was only introduced in 2020, and the very first cases performed with this novel technique were included in the present analysis. Another possible bias is the lack of strict criteria to select patients for the SL approach; at our center, all the patients harboring a cT1-2N0M0 renal mass suitable for NSS are offered an ocRAPN, and the indication for RR is currently decided during surgery, in case of profuse incoercible bleeding or iatrogenic injury of the collecting system. Consequently, our control group also included patients who were not theoretically eligible for both approaches but strictly necessitated the RR.

## 5. Conclusions

The patients in our study who, after 2020, received sutureless RAPN without hilum dissection and clamping, showed a good safety profile, low complications, and reduced incidence of perioperative and short-term renal function decrease. In our hands, and in the cases shown in this study, the sutureless technique, associated with monopolar coagulation of the resected bed, was associated with a statistically significant increase in trifecta achievement, as compared to standard renorrhaphy cases. Although our results are encouraging and stimulate further investigation, a larger study group, patients’ randomization, renal scan-based assessment of renal function, and longer follow-up are required before drawing definite conclusions on the potential long-term benefit of our proposed technique in kidney cancer surgery.

## Figures and Tables

**Figure 1 cancers-15-00698-f001:**
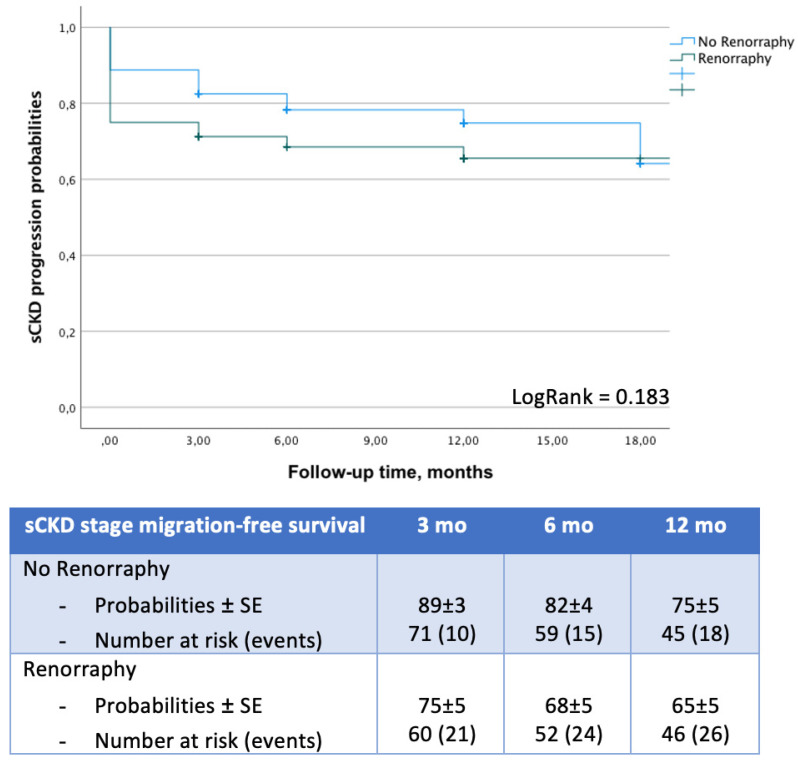
Impact of renorrhaphy on significant CKD stage migration-free survival (SMFS) probabilities after ocRPN.

**Table 1 cancers-15-00698-t001:** Patients’ characteristics and outcomes after ocRPN.

	Overall *n* = 533	SL *n* = 175 (33%)	RR *n* = 358 (67%)	*p*	PSM SL *n* = 80	PSM RR *n* = 80	*p*
Age, years	62 (54–71)	64 (55–72)	61 (52–70)	0.04	64 (56–73)	64 (55–73)	0.84
Male gender, *n* (%)	333 (62%)	108 (62%)	225 (63%)	0.79	51 (64%)	53 (66%)	0.74
ASA ≧ 3, *n* (%)	124 (23%)	32 (18%)	92 (26%)	0.06	13 (16%)	21 (26%)	0.12
Baseline eGFR, ml/min	81.8 (69.4–95.7)	82.1 (68.3–97.2)	81.7 (69.7–94.2)	0.59	79 (66.9–95.1)	78.1 (69.5–93.5)	0.86
Baseline CKD stage				0.76			0.74
1–2	466 (87%)	155 (88%)	311 (87%)		71 (89%)	70 (88%)	
3A–3B	59 (11%)	17 (10%)	42 (12%)		7 (9%)	9 (11%)	
4–5	8 (2%)	3 (2%)	5 (1%)		2 (2%)	1 (1%)	
Tumor size, cm	3.5 (2.5–5)	3 (2.5–5)	4 (3–5)	0.16	3 (2.5–4.5)	3.5 (2.5–5)	0.70
cT2, *n* (%)	70 (13%)	20 (11%)	50 (14%)	0.41	6 (7%)	6 (7%)	1.000
RENAL score				0.005			1.000
≤6	192 (36%)	80 (46%)	112 (31%)		32 (40%)	32 (40%)	
7–9	228 (43%)	65 (37%)	163 (46%)		37 (46%)	37 (46%)	
≥10	113 (21%)	30 (17%)	83 (23%)		11 (14%)	11 (14%)	
LOS, d	3 (2–5)	2 (2–5)	3 (3–5)	<0.001	2 (2–3)	3 (3–4)	<0.001
Blood transfusions, *n* (%)	24 (4%)	6 (3%)	18 (5%)	0.40	2 (2%)	5 (6%)	0.25
Postoperative eGFR, ml/min	73.6 (58.9–87.7)	75.7 (60.9–90.9)	72.7 (58.1–86.7)	0.13	76.1 (62.7–89.5)	72.9 (56.3–84.1)	0.15
Postoperative CKD stage				0.61			0.12
1–2	397 (75%)	135 (77%)	262 (73%)		65 (81%)	54 (68%)	
3A–3B	120 (22%)	35 (20%)	85 (24%)		13 (17%)	24 (30%)	
4–5	16 (3%)	5 (3%)	11 (3%)		2 (2%)	2 (2%)	
*Trifecta*, *n* (%)	458 (86%)	163 (93%)	295 (83%)	<0.001	77 (96%)	67 (84%)	0.008
pSM, *n* (%)	8 (1.5%)	3 (1.7%)	5 (1.4%)	0.78	1 (1%)	1 (1%)	0.99
CD ≧ 3 complications, *n* (%)	21 (4%)	3 (2%)	18 (5%)	0.06	1 (1%)	5 (6%)	0.09
sRFD, *n* (%)	54 (10%)	10 (6%)	44 (12%)	0.02	2 (2%)	9 (11%)	0.03

SL = sutureless, RR = renorrhaphy, PSM = propensity score-matched analysis, ASA = American Society of Anesthesiologists score, eGFR = estimated glomerular filtration rate, LOS = length of stay, pSM = positive surgical margins, CD = Clavien–Dindo grade, and sRFD = significant renal function deterioration.

**Table 2 cancers-15-00698-t002:** Uni-/multivariable logistic regression analyses to identify predictors of trifecta nonachievement and Cox analyses to identify predictors of significant CKD stage migration.

	Univariable Logistic Regression Analysis to Identify Predictors of Trifecta Achievement				Multivariable Logistic Regression Analysis to Identify Predictors of Trifecta Achievement				Univariable Cox Analysis to Identify Predictors of Significant CDK Stage Migration				Multivariable Cox Analysis to Identify Predictors of Significant CDK Stage Migration			
	OR	95% CI		*p*	OR	95% CI		*p*	HR	95% CI		*p*	HR	95% CI		*p*
		Lower	Higher			Lower	Higher			Lower	Higher			Lower	Higher	
Age	**0.98**	**0.96**	**0.99**	**0.03**	**0.97**	**0.95**	**0.99**	**0.01**	**1.04**	**1.01**	**1.07**	**0.003**	**1.03**	**1.003**	**1.07**	**0.03**
Male gender	0.85	0.51	1.41	0.53	-	-	-	-	0.72	0.39	1.34	0.30	-	-	-	-
ASA ≧ 3	0.68	0.39	1.15	0.15	-	-	-	-	1.54	0.79	2.99	0.21	-	-	-	-
Pre-eGFR	1.02	1.02	1.02	<0.001	0.99	0.99	1.01	0.92	**0.98**	**0.97**	**0.99**	**0.006**	**0.99**	**0.97**	**0.99**	**0.05**
RENAL score				**<0.001**				**<0.001**				**0.65**	-	-	-	-
≤6	**ref**	**-**	**-**		**ref**	**-**	**-**		ref	-	-					
7–9	**0.68**	**0.37**	**1.26**	**0.22**	**0.67**	**0.35**	**1.27**	**0.22**	0.82	0.40	1.68	0.63				
≥10	**0.29**	**0.15**	**0.56**	**<0.001**	**0.29**	**0.15**	**0.57**	**<0.001**	0.40	0.53	2.66	0.39				
Renorrhaphy	**0.31**	**0.16**	**0.60**	**<0.001**	**0.34**	**0.17**	**0.67**	**0.002**	0.87	0.43	1.75	0.69	-	-	-	-

ASA = American Society of Anesthesiologists score. Pre-eGFR = preoperative estimated glomerular filtration rate.

## Data Availability

Data supporting reported results are deposited at https://gbox.garr.it/garrbox/index.php/s/w4XVqbXO7CgdwH4 accessed on 18 January 2023.

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
