# Peer review of "Sutureless Purely Off-Clamp Robot-Assisted Partial Nephrectomy: Avoiding Renorrhaphy Does Not Jeopardize Surgical and Functional Outcomes"

_cancers, 2023, doi:10.3390/cancers15030698_

Round 1

Reviewer 1 Report

Authors tested the impact of sutureless RAPN on both surgical and functional outcomes. I believe the manuscript should be intensively modified before being considered for acceptance. Comments:

-Authors should report results both before (unmatched) and after 1:1 PS matching

-Have authors tried to match in a 1:2 ratio?

-Since the definition of sRFD is based on CKD categories, authors should report baseline CKD categories and CKD categories after surgery for both groups.

-I believe the current analysis significantly impacted by selection bias. Despite authors tried to use the PS matching to reduce imbalances, RR group harbored bigger and more complex tumors. Indeed, after matching, only 80 cases in each group were available (big differences between groups). Last, authors compared their last cases (SL) to their prime cases (RR) and, in consequence, the impact of learning curve could have generated misleading results. I believe authors should significantly tone down their message and extensively discuss these limitations. Moreover, I’m not sure that authors could demonstrate the superiority of SL over RR, but only report feasibility in selected cases

-After matching, authors selected a population of “less complex” cases in both groups. It will be interesting to provide results of the most complex cases in the SL cohort. Please, therefore, report also results of the unmatched populations

-After matching there were still differences in ASA score between groups (10%; p=0.1). I don0t think that, in this case, PS matching performed correctly

-Again, since the big difference is trifecta rates is due to sRFD percentage, I believe that these results are related to selection bias (for example lower ASA in SL). Could the authors provide deltaeGFR for both groups (i.e. a continuously coded measure of acute kidney injury, rather than a categorization in CKD classes).

-It’s quite counterintuitive to see higher blood transfusions in the RR group. Could the authors provide median Hb drop and intraoperative blood loss for both groups. In my opinion, the results of sRFD could be also be biased by blood loss, which could not so different between groups (same off clamp enucleation technique). If there are major differences between groups, probably results are biased by more ancient and more complex cases treated in the RR group.

-Some errors In the results: line 154 reported 3 vs 4% in transfusion rates (in Table 1 is different).

Lines 158-159, PSM did not generate two cohorts of 138 patients and these patients were not homogeneous for ASA score.

Lines 160-161 reported Trifecta after PS matching (table 1 reported different rates)

Lines 161-162 p value for KM is different in the Figure

-I believe authors predicted “No trifecta” rather than “Trifecta” in Log reg models

-Authors reported the abbreviation for BMI in the Table 2. However BMI is never reported. Do authors have BMI information?

-I believe the multivariable models are significantly unadjusted for confounders. An OR 0.31 is probably too generous. Moreover, why Renorraphy was not tested in Multi Cox models?

-Detailed postoperative complications for both groups should be reported.

Author Response

Authors tested the impact of sutureless RAPN on both surgical and functional outcomes. I believe the manuscript should be intensively modified before being considered for acceptance.

Comments:

  1. Authors should report results both before (unmatched) and after 1:1 PS matching

We thank the reviewer for the suggestion.

We also noticed that, before PSM, patients in the SL cohort showed a significantly lower rate of sRFD (6% vs 12%; p=0.02) and, consequently, a higher Trifecta rate (93% vs 83%; p<0.001). However, these results might be explained by the baseline differences in age and tumor complexity. For that reason, although these outcomes are clearly shown in Table 1, we preferred not to highlight them in the manuscript.

Moreover, our study was not designed to assess the superiority of any of the two surgical techniques so that it may be wise not to deliver such message to the reader. Further randomized-controlled trials (RCTs) are awaited to investigate possible advantages of the SL approach compared to RR.

  1. Have authors tried to match in a 1:2 ratio?

We thank the reviewer for his/her suggestion.

We definitely tried a 1:2 matching ratio, but the software was not able to provide us with two cohorts matched for age, gender, ASA score, baseline eGFR and RENAL score (especially considering that we required an exact matching of tumor complexity in the two groups).

  1. Since the definition of sRFD is based on CKD categories, authors should report baseline CKD categories and CKD categories after surgery for both groups.

We thank the reviewer for the suggestion.

Actually, as already specified in the manuscript (Materials and Methods, Page 2, lines 82-88), sRFD was defined as “a >30% reduction in the post-operative eGFR”, according to the National Kidney Foundation recommendations, while we considered as sCKDsm “any worsening from stages I-II to ≧ IIIa, from IIIa to ≧ IIIb, from IIIb to ≧ IV (doi:10.1053/J.AJKD.2014.07.030.) (doi:10.23736/S0393-2249.19.03570-7.).

As suggested by the reviewer, we modified Table 1 showing baseline and postoperative CKD stages, both before and after PSM. As expected, no significant difference was observed between groups (all p>0.12).

  • (Table 1, Page 6):
  1. I believe the current analysis significantly impacted by selection bias. Despite authors tried to use the PS matching to reduce imbalances, RR group harbored bigger and more complex tumors. Indeed, after matching, only 80 cases in each group were available (big differences between groups). Last, authors compared their last cases (SL) to their prime cases (RR) and, in consequence, the impact of learning curve could have generated misleading results. I believe authors should significantly tone down their message and extensively discuss these limitations. Moreover, I’m not sure that authors could demonstrate the superiority of SL over RR, but only report feasibility in selected cases

We thank the reviewer for the comment.

We totally agree with him/her that the two groups were statistically different at baseline, mainly for age (p=0.04) and RENAL score (p=0.005) and these observations are clearly shown in Table 1.

As noticed by the reviewer, tumors allocated in the RR cohort were significantly more complex than those in the SL group. In fact, the latter also included patients from our SL-specific learning curve: therefore, it is not surprising that we started performing this novel approach with uncomplex tumors, as these are usually characterized by a smaller contact-surface area. On the other hand, the control group is mainly (but not exclusively) composed of patients treated between 2017 and January 2020: at that time, the learning curve of RR-ocRAPN had long past and our proficiency with that approach was at the plateau.

However, a propensity score matched analysis was performed to address the imbalances between the 2 study groups: after matching, two cohorts of 80 patients each were obtained and no difference was observed concerning tumor complexity (p=1.000).

As said, the present study was not designed to assess the superiority of any of the two techniques. Ours is just a hypothesis-generating study and no definitive conclusion can be drawn in the absence of RCTs. We clearly acknowledged its main limitations, which are mainly (but not exclusively) related to its retrospective design, at the end of the manuscript.

  • (Discussion, Page 9, lines 301-304): This study, however, suffers from the limitations inherent to its retrospective design: because of this, it should be considered as a hypothesis-generating study and results from a randomized controlled trial are awaited in order to assess possible advantages of any of the two techniques.
  1. After matching, authors selected a population of “less complex” cases in both groups. It will be interesting to provide results of the most complex cases in the SL cohort. Please, therefore, report also results of the unmatched populations

We thank the reviewer for the suggestion.

Actually, patients’ characteristics and post-RAPN results of the study population overall and of the two study cohorts before PSM are already reported in the left half of Table 1. These outcomes, however, were not stressed in the manuscript, deliberately. In fact, as said above, our study was not designed to assess the superiority of any of the two surgical techniques so that it may be wise not to deliver such message to the reader. Further randomized-controlled trials (RCTs) are awaited to investigate possible advantages of the SL approach compared to RR.

However, we totally agree with the Reviewer that most cases in the SL group were not complex in terms of RENAL score, and small (mainly cT1, only a few cT2). This, however, does not diminish the value of our paper as we are not trying to show that our technique is feasible in any case, but it is safe, effective and beneficial in selected patients.

This point was stressed in the discussion and acknowledged in the dedicated section at the end of the manuscript.

  • (Discussion, Page 8, lines 262-267) Most of patients in the SL group harbored small (3 cm; IQR: 2.5-5), uncomplex renal tumors (83% RENAL <10) and less than 1 out of five presented with an ASA score ≧ 3. When compared to patients from the RR cohort (whose baseline characteristics were slightly, though not significantly, worse) the former reported a higher Trifecta rate (93% vs 83%; p<0.001), and this difference was confirmed after PSM analysis (96% vs 84%; p=0.008) (Tab. 1).
  • (Discussion, Page 9, lines 304-307) Moreover, most of patients in the SL group harbored small (3 cm; IQR: 2.5-5), uncomplex renal tumors (83% RENAL <10) and less than 1 out of five presented with an ASA score ≧ 3: this significantly limits the generalizability of our findings.
  1. After matching there were still differences in ASA score between groups (10%; p=0.1). I don0t think that, in this case, PS matching performed correctly

We thank the reviewer for his/her comment.

In the study population overall, approximately one out of 4 patients presented with an ASA score ≧ 3. The share was different in the SL and RR groups, both before (18% vs 26%) and after PSM (16% vs 26%); the difference, however, was NOT statistically significant (both p>0.06).

In fact, as already specified in the manuscript (Statistical analysis, Page 4, lines 150-153), “a 1:1 propensity score matched (PSM) analysis was used to minimize imbalances between the two cohorts for variables potentially affecting outcomes (age, gender, ASA score, baseline eGFR, RENAL score): the model was set to provide a standardized mean difference ≦10% between covariates.”

  1. Again, since the big difference in trifecta rates is due to sRFD percentage, I believe that these results are related to selection bias (for example lower ASA in SL). Could the authors provide deltaeGFR for both groups (i.e. a continuously coded measure of acute kidney injury, rather than a categorization in CKD classes).

We thank the reviewer for his/her comment.

As shown in Table 1, renal function of patients in the SL and RR cohorts are comparable, both before and after RAPN, and also after PSM. This observation is also supported by our KM analysis which provided evidence that RR does not significantly affect SMFS (Figure 1, Page 7).

As required by the reviewer, we assessed the ΔeGFR which was -7.14 (-17,15/0) in the overall population, -6.49 (-14.7/0) and -7.99 (-18.3/0) in the SL and RR cohorts, respectively (p=0.09). After matching, ΔeGFR was -4.88 (-11.63/0) and -8.78 (-18.12/0) in the two groups (p=0.052). Again, no difference was observed between groups. Therefore, considering these data redundant, we decided not to report them neither in the manuscript nor in the tables.

Concluding, RR seems to have a limited impact on renal function as only the share of patients with a sRFD is statistically different in the two groups, both before and after matching. Only assessing pre- and post-operative renal function with renal scans during further RCTs may clarify if the SL approach actually provides an advantage in terms of GFR preservation.

  1. It’s quite counterintuitive to see higher blood transfusions in the RR group. Could the authors provide median Hb drop and intraoperative blood loss for both groups. In my opinion, the results of sRFD could also be biased by blood loss, which could not be so different between groups (same off clamp enucleation technique). If there are major differences between groups, probably results are biased by more ancient and more complex cases treated in the RR group.

We thank the reviewer for the comment.

As already reported in Table 1, the transfusion rate was 4% overall. Patients experiencing this complication were slightly more common in the RR group compared to the SL cohort (5% vs 3%), but the difference was not statistically significant (p=0.40). As expected, no significant difference was observed after PSM too (6% vs 2%; p=0.25).

Therefore, in our opinion, it is hard to relate the statistically significant difference observed between the two groups in terms of sRFD (both before and after PSM) with bleeding episodes.

In fact, as noticed by the reviewer, intraoperative bleeding mainly occurs during tumor enucleation, which was performed through a purely off-clamp approach in all the cases.

Unfortunately, we are unable to collect data concerning the hemoglobin drop for all the included patients. However, we already provided evidence that the off-clamp approach is safe and feasible also when dealing with large (doi: 10.1016/j.eururo.2018.05.004.) and complex renal masses (doi: 10.1111/iju.14763) and the slightly increased intraoperative bleeding (compared to a conventional on-clamp RAPN) does not increase the risk of blood transfusion.

  1. Some errors In the results:

Line 154 reported 3 vs 4% in transfusion rates (in Table 1 is different).

Lines 158-159, PSM did not generate two cohorts of 138 patients and these patients were not homogeneous for ASA score.

Lines 160-161 reported Trifecta after PS matching (table 1 reported different rates)

Lines 161-162 p value for KM is different in the Figure

We thank the reviewer for noticing the typos.

Data shown in Table 1 and 2 were checked and found accurate. Therefore, the manuscript was amended accordingly:

  • (Results; Page 4, lines 174-189): Major complications (2% vs 5%; p=0.06) and transfusion (3% vs 5%; p=0.40) rates were not statistically different in the two groups. Overall, 5 (1%) urinary fistulas at the level of the parenchymal breach were observed after surgery (3 and 2 in the SL and RR groups, respectively; p=0.18) and they all healed spontaneously after ureteral stent placement. Selective embolization under general anesthesia was required 16 times overall, because of post-RAPN bleeding.

The PSM analysis generated 2 cohorts of 80 patients each, homogeneous for age, gender, ASA score, baseline eGFR, tumor size and RENAL score (all p>0.12). Patients receiving SL had shorter hospital stay (p<0.001) and increased likelihood of achieving Trifecta (96% vs 84%, p=0.008) (Tab. 1). SMFS probabilities were comparable at KM analysis (Log Rank = 0.183) (Figure 1). Multivariable logistic regression analysis identified age (OR: 0.97; 95%CI: 0.95-0.99; p=0.01), RENAL score ≧ 10 (OR: 0.29; 95%CI: 0.15-0.57; p<0.001) and renorrhaphy (OR: 0.34; 95%CI: 0.17-0.67; p=0.002) as independent predictors of Trifecta non-achievement.

  1. I believe authors predicted “No trifecta” rather than “Trifecta” in Log reg models

We thank the reviewer for noticing the typo.

Through logistic regression models we actually assessed predictors of Trifecta non-achievement.

Both the manuscript and the title of Table 2 were modified accordingly.

  • (Statistical Analysis; Page 4, lines 160-161): Univariable and multivariable logistic regression analyses were performed to identify predictors of Trifecta non-achievement.
  • (Results; Page 4, lines 186-189): Multivariable logistic regression analysis identified age (OR: 0.97; 95%CI: 0.95-0.99; p=0.01), RENAL score ≧ 10 (OR: 0.29; 95%CI: 0.15-0.57; p<0.001) and renorrhaphy (OR: 0.34; 95%CI: 0.17-0.67; p=0.002) as independent predictors of Trifecta non-achievement.
  • (Table 2; Page 6; lines 212-213): Uni/-multivariable logistic regression analyses to identify predictors of Trifecta non-achievement and Cox analyses to identify predictors of significant CKD stage migration.
  1. Authors reported the abbreviation for BMI in the Table 2. However BMI is never reported. Do authors have BMI information?

We thank the reviewer for noticing the typo.

BMI was removed from the legend in Table 2.

We agree with him/her that assessing body mass index differences between patients in the SL and RR cohort is not relevant to the aims of the present paper as it should not affect outcomes of RAPN.

  1. I believe the multivariable models are significantly unadjusted for confounders. An OR 0.31 is probably too generous. Moreover, why Renorraphy was not tested in Multi Cox models?

We thank the reviewer for the comment.

As previously acknowledged, the main limitation of our study is its retrospective design: consequently, only few baseline characteristics of the included patients could be retrieved from their files. However, we believe that age, male gender, ASA score, preoperative renal function and tumor complexity are the main unmodifiable variables affecting outcomes of RAPN.

As suggested by the reviewer, both univariate Logistic and Cox regression analyses were performed again to confirm the accuracy of the results displayed in Table 2. These tests were used to identify modifiable and unmodifiable variables significantly associated with the investigated outcomes of ocRAPN (Trifecta non-achievement and sCKDsm, respectively), which were further computed in the respective multivariable models to point out independent predictors.

Since univariable Cox analysis found that RR was not statistically associated with the risk of sCKDsm, it was not included in the multivariable model, on purpose.

  1. Detailed postoperative complications for both groups should be reported.

We thank the reviewer for his/her suggestion.

Actually, as already shown in Table 1, the observed transfusion rate was 4% overall and it was not statistically different among patients in the SL and RR cohorts, both before (3% vs 5%; p=0.40) and after PSM (2% vs 6%; p=0.25). Also the rate of CD ≧ 3 complications was comparable in the two groups, both before (2% vs 5%; p=0.06) and after PSM (1% vs 6%; p=0.09).

More in detail, 21 major complications were observed overall (4%). Five of these (3 and 2 in the SL and RR groups, respectively; p=0.18) were urinary fistulas diagnosed after RAPN, which required stent placement under general anesthesia. The 16 remaining were post-operative bleedings which required selective embolization: these all occurred in the RR group.

  • (Results; Page 4; lines 174-179): Major complications (2% vs 5%; p=0.06) and transfusion (3% vs 5%; p=0.40) rates were not statistically different in the two groups. Overall, 5 (1%) urinary fistulas at the level of the parenchymal breach were observed after surgery (3 and 2 in the SL and RR groups, respectively; p=0.18) and they all healed spontaneously after ureteral stent placement. Selective embolization under general anesthesia was required 16 times overall, because of post-RAPN bleeding.

Reviewer 2 Report

It is a very good idea and I think it is an analysis of good surgical results.

I have some queries:

1. Don't the authors do hilar dissection even for large tumors(T1b)? and what could you do in case of substantial bleeding when you were doing without hilar dissection?

2. What method is performed when the distinction between tumor and normal tissue is not clear?

I salute the authors for their innovative ideas and their wonderful experiences.

Author Response

It is a very good idea and I think it is an analysis of good surgical results.

I have some queries:

  1. Don't the authors do hilar dissection even for large tumors(T1b)? and what could you do in case of substantial bleeding when you were doing without hilar dissection?

We thank the reviewer for the questions.

As reported in the dedicated section, “Dissection of the renal hilum was not attempted as in no case were the vessels clamped. In fact, an hilum-sparing approach helps reducing surgical times while limiting accidental vascular injuries and, ultimately, intraoperative blood loss” (Materials and methods, Page 3, lines 119-122)

Preparing renal artery and vein is a time-consuming surgical step, which is also potentially associated with the risk of vascular injuries and major intraoperative bleeding. While it is obviously required during radical nephrectomy, it can be avoided during partial nephrectomy (even when dealing with cT2 renal masses [10.3390/cancers14184431]). In fact, in our institution, no need for intraoperative conversion to radical nephrectomy was recorded in the last two decades. Even more interestingly, we provided evidence that also in the hands of a trainee, hilar clamping is never required and it is safe to approach robot-assisted radical nephrectomy directly with a purely off-clamp technique [10.23736/S2724-6051.20.03673-5.].

In open surgery, intraoperative bleeding can be easily controlled compressing the kidney parenkyma. When a minimally invasive approach is preferred, it is crucial to progressively control all the feeding arteries identified during the enucleation, although a compete hemostasis is only pursued once tumor is excised (either with renorrhaphy or monopolar coagulation of the tumor bed). Also in case of relevant bleeding, the combination of irrigation and suction obtained resorting to two suction-irrigation cannulas ensures a clear surgical field. Moreover, by gently compressing major bleeding vessels with these cannulas, the table-side assistant is able to minimize intraoperative blood loss.

  1. What method is performed when the distinction between tumor and normal tissue is not clear?

We thank the reviewer for the questions.

As reported in the dedicated section, tumor enucleation was always “carried on with blunt dissection, following an almost avascular plane, while the two suction-irrigation devices helped maintaining a clear surgical field combining irrigation and suction. Feeding arteries identified during the enucleation were progressively controlled with monopolar coagulation […]” (Materials and methods, Page 3, lines 126-129). Thanks to this approach, we are always able to clearly identify and follow the proper enucleation plane, so that an enucleo-resection technique is only performed when protrusions of the tumor in the healthy parenchyma are suspected.

Reviewer 3 Report

The authors present an analysis of their consecutive series of robotic partial nephrectomy done by enucleation technique without hilar clamping or renorraphy. This is an extension of prior work where feasibility of this approach was seen with tumors with low/intermediate nephrometry scores. They extend this work to include some high nephrometry/complex tumors (n=30).  They hypothesize that avoiding deep sutures may result in improved renal function, an acceptable complication rate, and an acceptable positive surgical margin rate. To address this hypothesis, the perform a propensity score matched analysis using patients who underwent traditional renorrhaphy. The results support the hypothesis.

The authors acknowledge multiple shortcomings that ultimately render this paper as only hypothesis-generating. This includes all the biases inherent in any clinical study short of a randomized clinical trial. Most concerning is an imbalance of tumor complexity: the sutureless group tended to have more simple tumors. However, this difference appears to be resolved by propensity score matching. Second, those kidneys that had to be converted from sutureless to renorraphy intraoperatively (due to bleeding or collecting system injury) were excluded from analysis. This goes against the usual convention of intention-to-treat analysis. Third, the sutureless group is from a recent series of consecutive patients. So the renorrhaphy patients are more like historical controls. Use of historical controls introduces more bias as surgical techniques may evolve over time.

The manuscript would benefit from a better description of the analysis of surgical margins. The authors describe an enucleation technique for all tumors.  It is difficult to believe that the positive surgical margin rate was only 0-1%.

Nonetheless, the results further support the possibility of avoiding deep sutures in many patients. Intraoperative judgement is still necessary in cases with collecting system injury or uncontrolled bleeding.

Minor note: PSM abbreviation used for two different meanings

Author Response

The authors present an analysis of their consecutive series of robotic partial nephrectomy done by enucleation technique without hilar clamping or renorrhaphy. This is an extension of prior work where feasibility of this approach was seen with tumors with low/intermediate nephrometry scores. They extend this work to include some high nephrometry/complex tumors (n=30).  They hypothesize that avoiding deep sutures may result in improved renal function, an acceptable complication rate, and an acceptable positive surgical margin rate. To address this hypothesis, they perform a propensity score matched analysis using patients who underwent traditional renorrhaphy. The results support the hypothesis.

  1. The authors acknowledge multiple shortcomings that ultimately render this paper as only hypothesis-generating. This includes all the biases inherent in any clinical study short of a randomized clinical trial. Most concerning is an imbalance of tumor complexity: the sutureless group tended to have more simple tumors. However, this difference appears to be resolved by propensity score matching. Second, those kidneys that had to be converted from sutureless to renorraphy intraoperatively (due to bleeding or collecting system injury) were excluded from analysis. This goes against the usual convention of intention-to-treat analysis. Third, the sutureless group is from a recent series of consecutive patients. So the renorrhaphy patients are more like historical controls. Use of historical controls introduces more bias as surgical techniques may evolve over time.

We thank the reviewer for the comments.

We totally agree with him/her that ours is just a hypothesis-generating study and no definitive conclusion can be drawn in the absence of randomized-controlled trials. We clearly acknowledged its main limitations, which are mainly (but not exclusively) related to its retrospective design, at the end of the manuscript.

  • (Discussion, Page 9, lines 301-304): This study, however, suffers from the limitations inherent to its retrospective design: because of this, it should be considered as a hypothesis-generating study and results from a randomized controlled trial are awaited in order to assess possible advantages of any of the two techniques.

As noticed by the reviewer, tumors allocated in the RR cohort were significantly more complex than those in the SL group. In fact, the latter also included patients from our SL-specific learning curve: therefore, it is not surprising that we started performing this novel approach with uncomplex tumors, as these are usually characterized by a smaller contact-surface area. On the other hand, the control group is mainly (but not exclusively) composed of patients treated between 2017 and January 2020: at that time, the learning curve of RR-ocRAPN had long past and our proficiency with that approach was at the plateau. Indeed this represent a limitation of the present study, and it was acknowledged:

  • (Discussion, Page 9, lines 308-313) Another limitation of the present study is that the two groups are not homogeneous in terms of surgeon’s experience with the two techniques. In fact, in 2017, G.S. had already completed his learning curve with the conventional RR-ocRAPN; on the contrary, the SL approach was only introduced in 2020 and the very first cases performed with this novel technique were included in the present analysis.

However, a propensity score matched analysis was performed to address the imbalances between the 2 study groups: after matching, two cohorts of 80 patients each were obtained and no difference was observed concerning tumor complexity (p=1.000). Our analysis confirmed a significant advantage of the SL approach in terms of Trifecta rate (96% vs 84%; p=0.008) and sRFD (2% vs 11%; p=0.03).

Patients that required intraoperative conversion from SL to RR were NOT excluded from the analysis but included in the latter cohort (as suggested by the reviewer). Consequently, our control group includes (at least 10/358) patients that were not theoretically eligible for both the approaches but strictly necessitated the RR: this may represent another (acknowledged) limitation of our study design.

We better clarified this point in the manuscript.

  • (Results, Page 4, lines 165-174): Out of 548 cases of ocRAPN, 15 were lost to follow-up and excluded from the analysis. Among the remaining 533 patients, 185 were offered a SL approach but 10 (5%) required conversion to RR due to intraoperative suspicion of urinary calyx injury (n=8) or uncontrollable bleeding (n=2): therefore, they were ultimately allocated in the RR group (Supplementary material, Table S1). Overall, the median age of our study population was 62 years (IQR: 54-71) and the average tumor size 3.5 cm (IQR: 2.7-5). The observed Trifecta rate was 86% (Table 1). Compared to those in the RR group (n=358), patients from the SL cohort were significantly older (64 years vs 61 years; p=0.04) and less frequently harbored highly complex neoplasms (RENAL score ≥ 10) (17% vs 23%; p=0.005) (Tab. 1).
  • (Discussion, Page 9, lines 313-318): Another possible bias is the lack of strict criteria to select patients for the SL approach: at our center, all the patients harboring a cT1-2N0M0 renal mass suitable for NSS are offered an ocRAPN, and the indication for RR is currently decided during surgery, in case of profuse incoercible bleeding or iatrogenic injury of the collecting system. Consequently, our control group also included patients that were not theoretically eligible for both the approaches but strictly necessitated the RR.
  • (Supplementary Material):

Please notice that no further analysis was performed on this subgroup of patients (e.g.: assessment of variables predicting the risk of conversion from SL to RR) as their number was too limited to draw any conclusion.

  1. The manuscript would benefit from a better description of the analysis of surgical margins. The authors describe an enucleation technique for all tumors.  It is difficult to believe that the positive surgical margin rate was only 0-1%.

We thank the reviewer for his/her comment.

We asked our data-managers to go through patients’ files again and check the surgical margin status: overall, 8 cases were found (3 in the SL cohort). Datasets were updated accordingly, and statistical analysis performed.

Updated findings are shown in Table 1. Our definitive pSM rate was 1.4% overall and no difference was observed between the SL and RR groups (1.7% vs 1.4%; p=0.777), also after propensity score matching (1% vs 1%; p=0.993).

These results are in line with a recent systematic review and metanalysis (based on 19 studies and 3551 patients) comparing outcomes of robot-assisted and open partial nephrectomy, where Xia et al. described a pSM rate that ranged between 0-23.9% (10.1089/end.2016.0351).

Please notice that the definitive Trifecta rate was not modified as all the patients that were diagnosed with a pSM also had at least another reason not to achieve the composite outcome.

  1. Nonetheless, the results further support the possibility of avoiding deep sutures in many patients. Intraoperative judgement is still necessary in cases with collecting system injury or uncontrolled bleeding.

We thank the reviewer for the comment.

Actually, at the end of our manuscript, we clearly acknowledged “the lack of strict criteria to select patients for a SL approach”. In fact, “at our center, all the patients harboring a cT1-2N0M0 renal mass are offered an ocRAPN and the indication for RR is eventually decided intraoperatively, in case of profuse incoercible bleeding or iatrogenic injury of the collecting system” (Discussion, Page 8, lines 226-271).

Even though we agree with the reviewer that it represents a significant bias of the present study, we also believe that it is not an actual limitation of the sutureless approach itself. In fact, also other surgical maneuvers to control post-enucleation bleeding (such as the use of topical hemostatic agents...) are only accomplished, if required, based on the intraoperative judgement of the treating urologist.

Indeed, the aim of the present study was not to prove that all the cT1-2 renal tumors may undergo a SL-ocRAPN: this approach, however, can be routinely attempted and the decision to perform a RR may be easily taken intraoperatively, at the time of the final check of the enucleation bed, without jeopardizing surgical and functional outcomes.

  1. Minor note: PSM abbreviation used for two different meanings

We thank the reviewer for the comment.

Disambiguation was required indeed. Therefore, PSM was kept for “propensity score matched analysis” while pSM was used ad abbreviation for “positive surgical margins”.

Reviewer 4 Report

This manuscript is interesting and offers some new data and interesting insights in the field of surgical resection technique modifications of confined kidney cancers, in order to prevent post-operative loss of kidney function. However, it needs a number of changes before being accepted for publication in Cancers. The changes needed are detailed in the text below.

TITLE: The most important finding of this paper, which is also its main strength, is that the surgical technique proposed by the Authors offers some advantages over the conventional technique. The confrontation is between two groups of non-randomized patients, analyzed retrospectically: patients submitted to off-clamp RAPN+renorraphy and patients submitted to off-clamp RAPN with surgical coagulation of the resection bed, without renorraphy. In this particular setting, the personal data of the Authors shows that avoiding renorraphy is not only safe and effective, but can be also beneficial to the patients. This is the real finding of this study, and this fact should be clearly stated in the title of the paper, which is at the moment a bit generic and not very attention-catching. A proposed change for the title, which would make it make it more accurate and interesting, is:

SURGICAL AND SHORT/MID-TERM FUNCTIONAL OUTCOMES OF PURELY OFF-CLAMP ROBOT-ASSISTED PARTIAL NEPHRECTOMY: SUTURLESS TECHNIQUE PLUS MONOPOLAR CAUTHERIZATION OF THE RESECTION BED CAN BE SAFE, EFFECTIVE AND BENEFICIAL

SIMPLE SUMMARY: The aim of the Simple Summary is to explain concisely and in layman’s terms why the paper was started and how the findings may impact current clinical practice and be beneficial to patients. The Authors should re-phrase it and make it more self-explicatory and direct for the readers: state what this study is adding and why it is important.

ABSTRACT: The period of the study should be clearly stated in the Abstract: from 2017 to…? How long was the minimum follow up? This is a very crucial clinical information that cannot be missing. Moreover, state that the MEDIAN tumor size was 3.5 cm. How many where the cT1? How many the cT2? If the cT2 were only a small part pf the group, as it seems, then it is inaccurate to state that the technique was utilized in the cT1-2 renal masses. It was utilized in cT1 cases and a small group of selected cT2 cases. This information must be represented more accurately in the abstract, and then also in the paper’s main text.  

Abstract last line: Compared to RR, our experience seems to show that the SL approach significantly increased the probability of achieving the Trifecta in the observed group of cases.

INTRODUCTION:

Line 33-35: The Authors should be careful with the clinical accuracy of this statement. Does this statement imply that high-volume centers did not perform open partial nephrectomies in the last 10 years? The reduction of the rate of radical nephrectomies is a general phenomenon that is not only related to the increasing trend of use of robotic surgery, but it is related to improvement and spread of nephron-sparing surgery techniques in general, especially in high volume centers. The same principles of “safe” kidney resection are true for open, laparoscopic and robotic surgery.

MATERIALS AND METHODS

Line 93: “A transurethral urinary catheter was inserted and the patient was placed in an extended flank position”.

Line 104: “Dissection of the renal hilum was not attempted as in no case were the vessels clamped”. This fact is also true, I believe, for the renorraphy cases, this was the technique in all cases, wasn’t it? This concept is a very relevant technical point, that the Authors should stress better here and then in the Discussion of the paper. Avoiding dissection of the hilum reduces the surgical time, can reduce blood loss and accidental vascular injury to the hilum and to small additional veins or arteries. It is a relevant technical point that for sure is helpful in limiting damage to post-operative kidney function, in all cases. Hilum-sparing surgery is part of the technical advantages of the Authors’ proposed approach, and it should be well underlined in the text.

Line 113: Was the SwiftCOAG ever used in the renorraphy cases and an additional procedure? It is worth to comment a bit more on this device, and state for how many minutes was it applied and if there were any complications associated with it. This device seem to be a very crucial part of the technique. Did the Authors ever attempt to perform an operation without it, using a simple monopolar instrument to coagulate the resection bed? Would they speculate that this kind of surgery can be safely done without this device or do they deem it absolutely necessary for the successful outcome reported in their study group? Moreover, where coagulation patches or other materials with coagulation-enhancing effect used by the Authors?. Do they think these materials are important or necessary? What can the Authors advice to surgeons who do not have the SwiftCOAG device? Any speculation on possible alternatives?

Line 125: Study objective: The objective of this study is not only to present a novel surgical technique, but it is to report a personal experience of 170 cases where an innovative approach was used, comparing retrospectically the results with the conventional technique previously used

RESULTS:

Line 145: Of the initial 546 cases, 15 were lost at follow-up. Consequently 531 patients were included in the analysis”. Why in TABLE 1 there are 533 patients? The number in the text and in TABLE1 do not match, and should be corrected, in several parts.

Line 159: shorter hospital stay is generical, it should be specified that it is 2 vs 3 days.

Line 151: Patients with RENAL score <6 were highly significantly more frequent in the SL group as compared to the RR group, this is a very significant piece of information that it should be honestly underlined by the Authors. What they are trying to show here, it is not that their technique is the “panacea” in any case, but that it is safe, effective and beneficial in suitable cases. Stating that in their group most cases were not complex in terms of RENAL score, and small (cT1, only a few cT2) does not diminish the value of the paper, but at the same time is a much more honest way of representing the results. Moreover, it should be underlined that the group of patients in the RR group showed a higher trend towards high ASA score, although this did not reach, by a minimum,  statistical significance, probably for the small number of cases.

One point that should be underlined in the Results, and then discussed in the Discussion, is the gender difference regarding the impact of renal surgery on post-operative kidney function.

Gender-specific differences in kidney cancer have been investigated and well reviewed in Mancini M. et al, Gender-related approach to kidney cancer management: moving forward, IJMS, 2020, a citation that should be added to the references. Moreover, a very interesting recent publication showed that the effect on renal function of ureteral clamping during robotic radical cystectomy is gender-specific, being women more protected as compared to men (reference to add: Ishiyama Y, et al. Association between ureteral clamping time and acute kidney injury during robot-assisted radical cystectomy, Current Oncology, 2021). The Authors should comment on these important issues in the Results and Discussion of the paper. Did they observe any difference in terms of response to renal injury in the male and female population? Can they identify any protective effect in the female group? Was the renorraphy technique more impacting in men as compared to females in the renorraphy group? If this was the case, then their proposed sutureless technique should be particularly indicated in male patients, where the loss of renal function can be more pronounced, as shown in previous experiences.

DISCUSSION:

Line 168: “An average 20% decline in function after NSS is observed in treated kidney”. Treated how? Surgically treated? At what time point is this observed? How does this compare to the Authors results?

What is the minimum follow up of the patients observed in the Authors experience?  This is a crucial clinical point and this piece of information is missing in this study. All the conclusions should be referred to the minimum follow up, and the Authors should talk about short/mid-term results and not to results in general.

Line 190: RR may be responsible for 2/3 of post-operative surgical loss: this is a hypothesis and should be presented as such by the Authors.

Line 216: Eleven cases, though, (28%) necessitated intraoperative repair of the collective system.

Line 228: From 2020, all those presenting with a cT1-2N0M0 (this is not accurate, given the small percentage of cT2 cases; it should be said all cT1 and some cT2 cases).

Line 229: 10/180 cases required RR because of an intraoperative suspicion of iatrogenic renal calyx injury (wasn’t it also for uncontrollable bleeding? How many patients in each case?).

Line 252: Another limitation of this study is that the two groups are not homogenous in terms of age, RENAL score, timing of surgery (before 2020 and after) and experience of the surgeons involved, as the SL group started to exist after 2020, while the other group was operated before that time. It should be also more clearly stated that the follow up was very short in a number of cases (how short? How many cases?) and no conclusions can be drawn on long-term renal function.

Line 2060-261: Re-phase more clearly, or erase the last sentence.

CONCLUSIONS

Re-phrase the Conclusions more clearly and accurately. For example:

“The patients in our study who, after 2020, received sutureless RAPN without hilum dissection and clamping, showed a good safety profile, low complications and reduced incidence of perioperative and short-term renal function decrease. In our hands, and in the cases shown in this study, sutureless technique, associated with monopolar coagulation of the resected bed with the SwiftCOAG coagulation device, was associated with statistically significant increase in the Trifecta achievement, as compared to standard renorraphy cases. Although our results are encouraging and stimulating further investigation, a larger study group, patients’ randomization and longer follow up are required before drawing definite conclusions on the potential long-term benefit of our proposed technique in kidney cancer surgery”.

Author Response

This manuscript is interesting and offers some new data and interesting insights in the field of surgical resection technique modifications of confined kidney cancers, in order to prevent post-operative loss of kidney function. However, it needs a number of changes before being accepted for publication in Cancers.

The changes needed are detailed in the text below.

  1. TITLE: The most important finding of this paper, which is also its main strength, is that the surgical technique proposed by the Authors offers some advantages over the conventional technique. The confrontation is between two groups of non-randomized patients, analyzed retrospectically: patients submitted to off-clamp RAPN+renorraphy and patients submitted to off-clamp RAPN with surgical coagulation of the resection bed, without renorraphy. In this particular setting, the personal data of the Authors shows that avoiding renorraphy is not only safe and effective, but can be also beneficial to the patients. This is the real finding of this study, and this fact should be clearly stated in the title of the paper, which is at the moment a bit generic and not very attention-catching. A proposed change for the title, which would make it make it more accurate and interesting, is: SURGICAL AND SHORT/MID-TERM FUNCTIONAL OUTCOMES OF PURELY OFF-CLAMP ROBOT-ASSISTED PARTIAL NEPHRECTOMY: SUTURLESS TECHNIQUE PLUS MONOPOLAR CAUTHERIZATION OF THE RESECTION BED CAN BE SAFE, EFFECTIVE AND BENEFICIAL

We thank the reviewer for the comment.

We totally agree with him/her that our novel approach could provide some advantages over the conventional one, but the present study must be only considered as hypothesis-generating; in fact, its retrospective design and the lack of a fine assessment of pre-/post-operative renal function (e.g.: resorting to renograms) does not allow the reader to draw enthusiastic conclusions.

Therefore, as suggested by Reviewer #1, we prefer toning down the message and wait for further RCTs to come.

  1. SIMPLE SUMMARY: The aim of the Simple Summary is to explain concisely and in layman’s terms why the paper was started and how the findings may impact current clinical practice and be beneficial to patients. The Authors should re-phrase it and make it more self-explicatory and direct for the readers: state what this study is adding and why it is important.

We thank the reviewer for the suggestion.

Accordingly, the simple summary was re-phrased to concisely explain to a layman reader the main findings of our study.

  • (Simple Summary, Page 2, lines 14-16): suturing the kidney after tumor excision can be omitted most of the times, without increasing the risks of complications or jeopardizing renal function
  1. ABSTRACT: The period of the study should be clearly stated in the Abstract: from 2017 to…? How long was the minimum follow up? This is a very crucial clinical information that cannot be missing. Moreover, state that the MEDIAN tumor size was 3.5 cm. How many where the cT1? How many the cT2? If the cT2 were only a small part pf the group, as it seems, then it is inaccurate to state that the technique was utilized in the cT1-2 renal masses. It was utilized in cT1 cases and a small group of selected cT2 cases. This information must be represented more accurately in the abstract, and then also in the paper’s main text.  
  2. Abstract last line: Compared to RR, our experience seems to show that the SL approach significantly increased the probability of achieving the Trifecta in the observed group of cases.

We thank the reviewer for his/her suggestions.

The abstract was modified accordingly:

  • (Abstract, Page 1, lines 17-40): to compare outcomes of sutureless (SL) vs renorrhaphy (RR) off-clamp robotic partial nephrectomy (ocRPN), we retrospectively analyzed procedures performed at our center, from January 2017 to April 2021, for cT1-2N0M0 renal masses. All the patients with a minimum follow-up < 1 month were excluded from the analysis. […] Overall, 531 patients were included, with a median tumor size of 3.5 cm (IQR: 2.7-5); 70 (13%) presented with a cT2 mass. […] Compared to RR, our experience seems to show that the SL approach significantly increased the probabilities of achieving the Trifecta in the observed group of cases.
  1. INTRODUCTION: Line 33-35: The Authors should be careful with the clinical accuracy of this statement. Does this statement imply that high-volume centers did not perform open partial nephrectomies in the last 10 years? The reduction of the rate of radical nephrectomies is a general phenomenon that is not only related to the increasing trend of use of robotic surgery, but it is related to improvement and spread of nephron-sparing surgery techniques in general, especially in high volume centers. The same principles of “safe” kidney resection are true for open, laparoscopic and robotic surgery.

We thank the reviewer for both the comment and the suggestion.

The first sentence of the introduction was re-phrased accordingly:

  • (Introduction, Page 2, lines 45-48): Over the last decades, the improvement and spread of nephron-sparing surgery (NSS) techniques, together with the increasing trend in the use of robot-assisted partial nephrectomy (RAPN) for the treatment of renal masses led to the adoption of radical nephrectomy for cT1 renal masses to less than 10%, especially in high volume centers
  1. MATERIALS AND METHODS

Line 93: “A transurethral urinary catheter was inserted and the patient was placed in an extended flank position”.

We thank the reviewer for the suggestion.

The manuscript was modified accordingly:

  • (Materials and methods, Page 3, lines 107-108): A transurethral urinary catheter was inserted and the patient was placed in an extended flank position.

Line 104: “Dissection of the renal hilum was not attempted as in no case were the vessels clamped”. This fact is also true, I believe, for the renorraphy cases, this was the technique in all cases, wasn’t it? This concept is a very relevant technical point, that the Authors should stress better here and then in the Discussion of the paper. Avoiding dissection of the hilum reduces the surgical time, can reduce blood loss and accidental vascular injury to the hilum and to small additional veins or arteries. It is a relevant technical point that for sure is helpful in limiting damage to post-operative kidney function, in all cases. Hilum-sparing surgery is part of the technical advantages of the Authors’ proposed approach, and it should be well underlined in the text.

We thank the reviewer for both the comment and the suggestion.

The manuscript was modified as follows:

  • (Materials and methods, Page 3, lines 119-120): Dissection of the renal hilum was not attempted as in no case were the vessels clamped. In fact, an hilum-sparing approach helps reducing surgical times while limiting accidental vascular injuries and, ultimately, intraoperative blood loss.

Line 113: Was the SwiftCOAG ever used in the renorraphy cases and an additional procedure? It is worth to comment a bit more on this device, and state for how many minutes was it applied and if there were any complications associated with it. This device seem to be a very crucial part of the technique. Did the Authors ever attempt to perform an operation without it, using a simple monopolar instrument to coagulate the resection bed? Would they speculate that this kind of surgery can be safely done without this device or do they deem it absolutely necessary for the successful outcome reported in their study group? Moreover, where coagulation patches or other materials with coagulation-enhancing effect used by the Authors?. Do they think these materials are important or necessary? What can the Authors advice to surgeons who do not have the SwiftCOAG device? Any speculation on possible alternatives?

We thank the reviewer for his/her comments and questions.

First of all, we believe it is worth clarifying that the SwiftCOAG is not a device itself, but one of the possible monopolar standard current waveforms that the DaVinci Xi built-in generator (ERBE VIO dV 2.0) can provide.

It was obviously used in all the 533 procedures, during the dissection of the tumor from the Gerota’s fat and at the time of enucleation. However, only in the case of a SL approach it was generously employed in order to achieve the hemostasis, after tumor excision.

Although our experience is limited to the robotic approach (so that an ERBE VIO generator was always available), also small laparoscopic series were previously published with a similar approach, and different generators and current waveforms were used (10.1111/iju.12276).

Second, we specified that topical hemostatic agents were never used in our series: in fact, we are convinced that these devices may conceal a bleeding vessel at the time of hemostasis-check, potentially resulting in a significant post-operative bleeding.

  • (Materials and methods, Page 3, lines 128-136):

Feeding arteries identified during the enucleation were progressively controlled with monopolar coagulation, but complete hemostasis was only pursued once tumor was excised: then, monopolar cautery (SwiftCOAG™ waveform, effect 8, 80 W limit) was extensively applied to the resection bed, under appropriate dripping of saline solution, while blood was aspirated. Once a firm uniform eschar covered the entire resection bed and no bleeders were observed, the pneumoperitoneum was lowered down to 5 mmHg and hemostasis was checked for 5 minutes. Topical hemostatic agents were never used to improve bleeding control. An EndoCatch device was used to retrieve the specimen.

Line 125: Study objective: The objective of this study is not only to present a novel surgical technique, but it is to report a personal experience of 170 cases where an innovative approach was used, comparing retrospectically the results with the conventional technique previously used

We thank again the reviewer for his/her suggestion.

The manuscript was modified accordingly:

  • (Study objective, Page3, lines143-145) The aim of the present study was to present our novel SL surgical technique and report our single-center experience of 175 cases, retrospectively comparing outcomes with those of the conventional ocRAPN with RR.
  1. RESULTS:

Line 145: Of the initial 546 cases, 15 were lost at follow-up. Consequently 531 patients were included in the analysis”. Why in TABLE 1 there are 533 patients? The number in the text and in TABLE1 do not match, and should be corrected, in several parts.

We thank the reviewer for noticing the typos.

Numbers reported in Table 1 were checked and found accurate. Therefore, the manuscript was amended accordingly:

  • (Results, Page 4, lines 165-169): Out of 548 cases of ocRAPN, 15 were lost to follow-up and excluded from the analysis. Among the remaining 533 patients, 185 were offered a SL approach but 10 (5%) required conversion to RR due to intraoperative suspicion of urinary calyx injury (n=8) or uncontrollable bleeding (n=2): therefore, they were ultimately allocated in the RR group (Supplementary material, Table S1).

Line 159: shorter hospital stay is generical, it should be specified that it is 2 vs 3 days.

We thank the reviewer for the suggestion.

The median number of hospital stay was also reported in the manuscript.

  • (Results, Page 4, lines 183-184): Patients receiving SL had shorter hospital stay (2 days vs 3 days; p<0.001)[…]

Line 151: Patients with RENAL score <6 were highly significantly more frequent in the SL group as compared to the RR group, this is a very significant piece of information that it should be honestly underlined by the Authors. What they are trying to show here, it is not that their technique is the “panacea” in any case, but that it is safe, effective and beneficial in suitable cases. Stating that in their group most cases were not complex in terms of RENAL score, and small (cT1, only a few cT2) does not diminish the value of the paper, but at the same time is a much more honest way of representing the results. Moreover, it should be underlined that the group of patients in the RR group showed a higher trend towards high ASA score, although this did not reach, by a minimum, statistical significance, probably for the small number of cases.

We thank the reviewer for the suggestion.

We totally agree with him/her that most tumors treated with a SL approach were small and uncomplex and only a few of these patients presented wtih an ASA score ≧ 3.

This point was stressed in the discussion and acknowledged in the dedicated section at the end of the manuscript.

  • (Discussion, Page 8, lines 262-267) Most of patients in the SL group harbored small (3 cm; IQR: 2.5-5), uncomplex renal tumors (83% RENAL <10) and less than 1 out of five presented with an ASA score ≧ 3. When compared to patients from the RR cohort (whosebaseline characteristics were slightly, though not significantly, worse) the former reported a higher Trifecta rate (93% vs 83%; p<0.001), and this difference was confirmed after PSM analysis (96% vs 84%; p=0.008) (Tab. 1).
  • (Discussion, Page 8, lines 304-307) Moreover, most of patients in the SL group harbored small (3 cm; IQR: 2.5-5), uncomplex renal tumors (83% RENAL <10) and less than 1 out of five presented with an ASA score ≧ 3: this significantly limits the generalizability of our findings.

One point that should be underlined in the Results, and then discussed in the Discussion, is the gender difference regarding the impact of renal surgery on post-operative kidney function. Gender-specific differences in kidney cancer have been investigated and well reviewed in Mancini M. et al, Gender-related approach to kidney cancer management: moving forward, IJMS, 2020, a citation that should be added to the references. Moreover, a very interesting recent publication showed that the effect on renal function of ureteral clamping during robotic radical cystectomy is gender-specific, being women more protected as compared to men (reference to add: Ishiyama Y, et al. Association between ureteral clamping time and acute kidney injury during robot-assisted radical cystectomy, Current Oncology, 2021). The Authors should comment on these important issues in the Results and Discussion of the paper. Did they observe any difference in terms of response to renal injury in the male and female population? Can they identify any protective effect in the female group? Was the renorraphy technique more impacting in men as compared to females in the renorraphy group? If this was the case, then their proposed sutureless technique should be particularly indicated in male patients, where the loss of renal function can be more pronounced, as shown in previous experiences.

We thank the reviewer for the suggestions.

Although it was far from the aim of our study, we performed a logistic regression analysis to investigate predictors of sRFD and results were reported in Table S2, which will be part of the supplementary material.

Interestingly, age at surgery, RENAL score ≧ 10  and renorrhaphy were identified as independent predictors of sRFD; no association was found with gender.

We briefly reported these findings in the Results section and further commented on them in the Discussion: both the suggested articles were cited.

  • (Results, Page 4, lines 189-191) Age at surgery (OR: 1.07; 95%CI: 1.04-1.11; p<0.001), RENAL score ≧ 10 (OR: 4.96; 95%CI: 2.23-11.06; p<0.001) and RR (OR: 2.35; 95%CI: 1.12-4.94; p=0.02) independently predicted sRFD (Supplementary material, Table S2).
  • (Discussion, Page 8, lines 285-299) It is worth highlighting that RR was an independent predictor of the risk of sRFD (OR: 2.35; 95%CI: 1.12-4.94; p=0.02), together with age at surgery (OR: 1.07; 95%CI: 1.04-1.11; p<0.001) and RENAL score ≧ 10 (OR: 4.96; 95%CI: 2.23-11.06; p<0.001). These results are in line with the available literature: in fact, it was hypothesized that the reconstructive injury caused by stitching the parenchymal breach may contribute to the post-operative functional loss[7]. Moreover, it is well known that GFR decreases with age and older patients have reduced capability to cope with ischemic kidney damage[13]. Furthermore, there is strong evidence that highly complex tumors are associated with an increased contact surface area, which in turn requires more energy (in the case of a SL approach) or stitches (when RR is performed) to achieve the hemostasis[11], thus increasing the risk of damaging the remaining healthy parenchyma. Surprisingly, our Logistic regression analysis failed to prove any association between gender and the response to renal injury: in fact, there is evidence that women may be more protected against temporarily impaired renal circulation[41,42].
  1. DISCUSSION:

Line 168: “An average 20% decline in function after NSS is observed in treated kidney”. Treated how? Surgically treated? At what time point is this observed? How does this compare to the Authors results?

What is the minimum follow up of the patients observed in the Authors experience?  This is a crucial clinical point and this piece of information is missing in this study. All the conclusions should be referred to the minimum follow up, and the Authors should talk about short/mid-term results and not to results in general.

We thank the reviewer for the comment and questions.

In the first sentence of the Discussion section, we were commenting on the latest evidence available concerning post-opeartive renal function decline: in fact, the following sentence includes the “resected heathy parenchyma”, the “ischemia/reperfusion damage” and the “reconstructive injury” in the list of its possible cause.

However, we better clarified this point as follows:

  • (Discussion, Page 4, lines 196-197) An average 20% decline in function after NSS is observed in the treated kidney right after partial nephrectomy.

Line 190: RR may be responsible for 2/3 of post-operative surgical loss: this is a hypothesis and should be presented as such by the Authors.

We thank the reviewer for his/her suggestion.

The manuscript was modified accordingly:

  • (Discussion, Page 7, lines 220-221) This, however, is hypothesized to be responsible of 2/3 of the post-operative functional loss[7].

Line 216: Eleven cases, though, (28%) necessitated intraoperative repair of the collective system.

We thank the reviewer for the suggestion.

The manuscript was modified accordingly:

  • (Discussion, Page 8, lines 245-246) Eleven cases (28%), though, necessitated intraoperative repair of the collecting system.

Line 228: From 2020, all those presenting with a cT1-2N0M0 (this is not accurate, given the small percentage of cT2 cases; it should be said all cT1 and some cT2 cases).

Line 229: 10/180 cases required RR because of an intraoperative suspicion of iatrogenic renal calyx injury (wasn’t it also for uncontrollable bleeding? How many patients in each case?).

We thank the reviewer for his/her comments and suggestions.

Actually, from January 2020 onwards, all those presenting at our institution with an organ-confined renal tumor which was suitable for nephron sparing surgery were offered a SL-ocRAPN, regardless tumor size and complexity. Obviously, cT2 tumors were rarer than cT1 renal masses; and even rarer were large organ-confined neoplasm suitable for partial nephrectomy. Therefore, the small amount of cT2 tumors allocated in the SL cohort is non properly the result of a selection bias but the evidence of the rarity of these cases.

Of course, some patients (n=2; Table S1, Supplementary material) harboring a cT2 cancer that were initially offered a SL approach, intraoperatively required conversion to RR.

  • (Discussion, Page 8, lines 257-260) From January 2020, all those presenting with a cT1-2N0M0 renal mass which was suitable for ocRAPN were treated with a SL intent: 10/180 cases (5%) required RR because of an intraoperative suspicion of iatrogenic renal calyx injury (n=8) or uncontrollable bleeding (n=2).

The causes of conversion from SL to RR were also reported in the Results section

  • (Results, Page 4, lines 166-169) Among the remaining 533 patients, 185 were offered a SL approach but 10 (5%) required conversion to RR due to intraoperative suspicion of urinary calyx injury (n=8) or uncontrollable bleeding (n=2): therefore, they were ultimately allocated in the RR group (Supplementary material, Table S1).

Line 252: Another limitation of this study is that the two groups are not homogenous in terms of age, RENAL score, timing of surgery (before 2020 and after) and experience of the surgeons involved, as the SL group started to exist after 2020, while the other group was operated before that time. It should be also more clearly stated that the follow up was very short in a number of cases (how short? How many cases?) and no conclusions can be drawn on long-term renal function.

We thank the reviewer for the comments.

As suggested, we clearly reported in the result section the median length of the follow-up for the entire study population and for the two cohors.

  • (Results, Page 4, lines 180-181) Median follow-up time was 17 months (IQR: 8-30): 27 months (IQR: 14-36) in the RR cohort and 9 months (IQR: 6-14) in the SL group.

As the follow-up of patients from the SL cohort was too short, no conclusions can be drawn on long-term outcomes of our novel technique. This limitation was already acknowledged.

  • (Discussion, Page 9, lines 306-307) Furthermore, to evaluate potential benefits of this procedure on long-term outcomes, a longer follow-up is required.

We clarified that all the procedures were performed by the same experienced robotic surgeon

  • (Materials and Methods, Page 3, line 104) All the procedures were performed by one single experienced robotic surgeon (G.S.).

We acknowledged the limitation concerning the different experience of our main surgeon with the two techniques.

  • (Discussion, Page 9, lines 308-313) Another limitation of the present study is that the two groups are not homogeneous in terms of surgeon’s experience with the two techniques. In fact, in 2017, G.S. had already completed his learning curve with the conventional RR-ocRAPN; on the contrary, the SL approach was only introduced in 2020 and the very first cases performed with this novel technique were included in the present analysis.

Line 260-261: Re-phase more clearly, or erase the last sentence.

We thank the reviewer for the comments.

As suggested, the last sentence was re-phrased as follows:

  • (Discussion, Page 9, lines 317-319) Consequently, our control group also included patients that were not theoretically eligible for both the approaches but strictly necessitated the RR.
  1. CONCLUSIONS

Re-phrase the Conclusions more clearly and accurately. For example: “The patients in our study who, after 2020, received sutureless RAPN without hilum dissection and clamping, showed a good safety profile, low complications and reduced incidence of perioperative and short-term renal function decrease. In our hands, and in the cases shown in this study, sutureless technique, associated with monopolar coagulation of the resected bed with the SwiftCOAG coagulation device, was associated with statistically significant increase in the Trifecta achievement, as compared to standard renorraphy cases. Although our results are encouraging and stimulating further investigation, a larger study group, patients’ randomization and longer follow up are required before drawing definite conclusions on the potential long-term benefit of our proposed technique in kidney cancer surgery”.

We heartily thank the reviewer for his/her suggestion and support.

The study conclusions were re-phrased as recommended:

  • (Conclusions, Page 9, lines 321-330) The patients in our study who, after 2020, received sutureless RAPN without hilum dissection and clamping, showed a good safety profile, low complications and reduced incidence of perioperative and short-term renal function decrease. In our hands, and in the cases shown in this study, sutureless technique, associated with monopolar coagulation of the resected bed, was associated with statistically significant increase in the Trifecta achievement, as compared to standard renorraphy cases. Although our results are encouraging and stimulating further investigation, a larger study group, patients’ randomization, renal-scan-based assessment of renal function and longer follow up are required before drawing definite conclusions on the potential long-term benefit of our proposed technique in kidney cancer surgery.

Round 2

Reviewer 1 Report

I Would like to thank the authors for addressing all my comments

Reviewer 4 Report

The Authors did a good job in revising and improving the text. A more critical approach to results and conclusions is recommended to the Authors as a general suggestion, for future publications. Further research and deeper conceptualisation would be advisable in future and further studies on the topic